# Preliminary Analysis of the Ductility and Crack-Control Ability of Engineered Cementitious Composite with Superfine Sand and Polypropylene Fiber (SSPP-ECC)

**DOI:** 10.3390/ma13112609

**Published:** 2020-06-08

**Authors:** Zhiqing Zhu, Guojin Tan, Weiguang Zhang, Chunli Wu

**Affiliations:** 1College of Transportation, Jilin University, Changchun 130022, China; zhuzq19@mails.jlu.edu.cn (Z.Z.); tgj@jlu.edu.cn (G.T.); clwu@jlu.edu.cn (C.W.); 2School of Transportation, Southeast University, Nanjing 211189, China

**Keywords:** SSPP-ECC, superfine sand, ductility, crack control, factor analysis

## Abstract

Engineered cementitious composite (ECC) is a potential cement-based material with the abilities of large deformation and crack width control. However, ECC is difficult to popularize in many developing countries because the costs of silica sand and polyvinyl alcohol (PVA) fiber with a surface coating are too high for practical engineering. Therefore, we proposed an economical ECC with superfine river sand and polypropylene (PP) fiber (SSPP-ECC) to replace PVA fiber and silica sand. The SSPP-ECC proposed in this paper is a sustainable material using local material ingredients, which has considerable adaptability for large-scale engineering applications. The 16 groups of specimens were prepared through a factorial design method, curing for four-point bending tests. The bending strength, deflection, flexural modulus of elasticity, and crack width were measured and calculated during the test. The factor analysis of the test results shows that the contents of fiber and fly ash had significant effects on the ductility of SSPP-ECC with an extra combined effect at the same time, and a response surface model with high accuracy was fitted to predict the yield length of SSPP-ECC. The ductility of SSPP-ECC was positively related to its crack-control ability and it was shown that the crack width of SSPP-ECC increased significantly with a high content of superfine sand. This paper proposed a reasonable way to utilize superfine sand and provided the mix proportion of SSPP-ECC with characteristics of deformation hardening and multi-cracking, which may cater to the demands of many concrete components on ductility and crack resistance.

## 1. Introduction

Engineered cementitious composite (ECC) has been developed to address traditional concrete shortcomings of brittleness and cracking. Reasonable tailored ECCs display superior tensile strength and cracking toughness under bending or tension loads, which macroscopically presents as noticeable metal-like pseudo-tensile ductility [1]. With the characteristics of obvious strain hardening and multiple fine cracking, ECC could resist excessive structural deformation and the permeation of erosion factors [2,3], which are meaningful in the fields of concrete durability, safety, and sustainability.

The studies that have been done on ECC are largely concentrated on laboratory investigations. Collecting the experimental results of different researchers together, it has been summarized that the conventional ECC has a compressive strength of 20–95 MPa, first crack strength of 3–7 MPa, ultimate tensile strain of 1–8%, specific gravity of 0.95–2.3, and so on [4,5,6,7]. It has also been verified that ECC has significant advantages regarding energy dissipation and spalling resistance over traditional concrete materials when exerted upon with complex loading conditions, such as shearing and twisting [8,9]. In terms of durability, Li et al. [10] tested the freeze–thaw durability of non-air-entrained ECC specimens, concluding a durability factor of 10 for concrete compared to 100 for ECC. Moreover, ECC specimens exposed to environments with a tropical climate, chloride, or deicing salt have shown promising durability in laboratory tests [11,12,13], while plain concrete is susceptible to degradation in these conditions. However, despite these advantages, research is still far from achieving globally mature engineering applications of ECC. One of the reasons is that most studies were focused on classical PVA-ECC, while some restrictions on engineering applications were neglected. For example, the commonly used PVA fiber for ECC technically has a surface coating on the nanometer scale [4], which is costly or unavailable in many areas. Consequently, the PP (polypropylene) fiber, which has always been used in concrete modification, was studied by a few researchers in the form of PP-ECC. It should be noted that the tensile strength and modulus of PP fibers are lower than PVA fiber, and the bonding strength with the mortar matrix is weaker as well. Therefore, the tensile strength, cracking toughness, and ductility of PP-ECC are below that of classical PVA-ECC [14]. Nevertheless, it has been amply confirmed that PP-ECCs still display the material characteristics of obvious strain hardening and multiple cracking [15], which would easily cater to the demands of many concrete structures. For large-scale applications of deformable cementitious composites in practical constructions, PP-ECC has great potential. Zhou et al. [16] conducted an experimental study on the torsional behavior of fiber-reinforced cement (FRC) and ECC beams reinforced with glass fibre reinforced polymer (GFRP) bars. The results indicated that PP fibers could effectively inhibit the crack propagations and the spacing and width of the cracks were decreased. Krishnaraja et al. [17] tested the flexural performance of PP-ECC-reinforced concrete beams, whose results revealed that PP fiber with a 0.65% volume fraction hybridized with steel with a 1.35% volume fraction mix has reasonable flexural performance and notable displacement ductility. Ismail et al. [18] studied the performance of PP-ECC beam-column joints under reversed cyclic loading, concluding that PP-ECC joints have better cracking behavior and higher shear strength compared to a normal concrete joint. The key is the effective combination of PP-ECC and reinforced concrete, which greatly improves the energy dissipation and spalling resistance of structures [19,20].

In addition to the economy and convenience of PP fiber, whether the source of sand conforms to the availability of most construction projects was considered in this paper. As the only aggregate of ECC, the sand used by researchers is always silica sand with an average grain size of about 110 μm to achieve a better dimension fit with fibers in the matrix [21]. However, silica sand is expensive or even unavailable in many areas, which hinders actual engineering applications. By comparison, river sand has always been one of the most important sources of engineering fine aggregate, which is often divided into coarse sand, medium sand, and superfine sand according to the particle size. What should be paid attention to is that coarse sand and medium sand have been widely used in various cement-based materials, while superfine sand is piled up in large quantities because of the lack of consumption [22], which may cause environmental problems, such as river blockage [23], as shown in Figure 1. Consequently, some scholars have researched superfine sand concrete. The results showed that the proportion of superfine sand should be quite low, otherwise the workability and mechanical properties of the concrete will be greatly reduced due to the increased water demand of fresh concrete and the reduction of spacing between coarse aggregates [24,25,26]. Therefore, it has become a common phenomenon that coarse sand and medium sand resources are scarce, while the superfine sand is rarely used in concrete production in many areas.

ECC is an emerging functional cement-based composite, where it is recommended that silica sand with a particle size of not more than 0.25 mm, without coarse aggregate, should be used. However, the production of silica sand involves multiple processing, which is relatively complex and costly. Therefore, if superfine sand can be used instead of silica sand, it will be more conducive to the practical applications of ECC materials. There were many clean superfine sand resources in the area we located, with an average particle size of about 0.25 mm. Classical PVA-ECC has good deformation and crack-control ability, such that it may be suitable for some specific concrete structures. However, there are many components in road and bridge engineering, such as wet joints, expansion joints, and concrete covers, that demand deformation and crack-control abilities that do not need to be as good as PVA-ECC. Especially in some developing countries, the economy and local materials are also important factors to be considered in engineering construction projects. Regarding the raw materials of PVA-ECC, PVA fiber and silica sand have high costs and are not easy to obtain in many places. Therefore, we put forward the possibility of an engineered cementitious composite with superfine sand and PP fiber (SSPP-ECC) based on the local superfine sand resources, which is conducive to accelerating the application of ECC in practical constructions.

In this paper, the effects of the amounts of fiber, fly ash, sand, and water on the properties of SSPP-ECC were studied with a factorial design/factor analysis method. The ductility and crack-control ability were studied based on the bending test and statistical analysis to understand the key performances of SSPP-ECC. This work provides references based on objective data for engineering application and further research of SSPP-ECC, which contributes to the balance between economical, sustainability, and material performance.

## 2. Materials and Methods

### 2.1. Raw Materials and Specimen Preparation

The SSPP-ECC was composed of cementitious binders, superfine sand, water, and PP fiber, where the cementitious binders included the ordinary Portland cement 52.5R and F.II grade fly ash produced by Jilin Yatai Group (Jilin, China). Table 1 shows the main components and properties of cementitious binders. The short PP fiber produced by Beijing CTA Fiber Co. Ltd. (Beijing, China) was used as reinforcement, whose properties are shown in Table 2. The superfine sand was from a river’s edge in Jilin province of China with a fineness modulus of 1.1, a maximum particle size of 0.6 mm, an average particle size of 0.25 mm, and mud content of 0.95%. Figure 2 illustrates the grading curve of the superfine sand used in this paper. Furthermore, the polycarboxylate-based high-range water reducer (HRWR) with about 1.2‰ cement quality was used to improve the matrix flowability.

To prepare the specimen, the raw materials were transformed using the following mixing process: (1) first, the cementitious materials were stirred to a viscous state with a water–binder ratio of 0.26; (2) then, the fiber and HRWR were added and mixed well; and (3) finally, the superfine sand and remaining water were added and the mixture was stirred for at least 5 min to ensure a uniform fiber distribution. The final evenly mixed mixture was in the state of cream texture, as shown in Figure 3.

The fresh matrices were created by pouring the mixture into 15 mm × 100 mm × 400 mm thin plate molds and left to harden. Afterward, the specimens were removed from the molds after 48 h and were cured for 28 days in a standard concrete curing environment.

### 2.2. Experimental Procedure

Factorial design/factor analysis is a statistical method used to quickly find out the main influencing factors of the inspection indexes and to analyze what determines the inspection indexes by quantifying the effects of multiple factors and their interactions. In this paper, four factors—volume content of fiber (A), ratio of fly ash to cementitious binders (B), water–binder ratio (C), and sand–binder ratio (D)—were selected to control the mix proportion of SSPP-ECC. Based on a literature review and practical experience, as well as considering the difference of raw materials between SSPP-ECC and conventional PVA-ECC, the levels of each factor were set within a reasonable range. A 2^4^ factorial design was used in this study, where the details are shown in Table 3. In particular, the setting of the range levels of factor B considered the impact of fly ash on the basic strength of the matrix, while the setting of the range levels of factor D considered the tendency to increase the consumption of superfine sand.

Based on the 2^4^ factorial design, 16 groups of specimens were prepared according to the mix proportions in Table 4. After curing for 28 days, the specimens with dimensions of 15 mm × 100 mm × 400 mm were tested using a mechanical testing and simulation (MTS) device produced by Measure Test Simulate Company (America) for four-point bending, per the Chinese specification GB/T 15231-2008 [27], using the strain-control condition at an axial strain rate of 0.2 mm/min, as shown in Figure 4. The MTS carried a load cell with a force measurement of 0–10 kN and its system recorded the changes in load and deflection in the whole process. At the same time, the crack widths of the specimens in different states were collected using the ZBL-F103 crack observation instrument produced by Beijing Zhi Bo Lian Science & Technology Co. Ltd. (Beijing, China), whose measurement accuracy is 10 μm, as exhibited in Figure 5.

For ECC materials, the ductility and crack width control are the most important properties. During the four-point bending test, the ductility of ECC was mainly reflected in the deformation ability from the initial crack strength to the peak strength. Therefore, the difference between the deflection corresponding to the peak strength and the deflection corresponding to the initial crack strength was used to quantify the ductility of SSPP-ECC by us, which was defined as the yield length in this study. Meanwhile, the maximum crack widths during the loading process were recorded. After the test, the initial crack strength, peak strength, yield length, and flexural modulus of elasticity of the 16 groups of specimens were calculated using Equations (1)–(4), respectively, according to Chinese specification GB/T 15231-2008 [27]: σ_1_ = P_1_L/bh^2^(1)
σ_m_ = P_m_L/bh^2^(2)
L_y_ = δ_m_ − δ_1_(3)
E = 23P_1_L^3^/162δbh^3^(4)
in which, σ_1_, σ_m_, L_y_, and E are the initial crack strength, peak strength, yield length, and flexural modulus of elasticity, respectively. P_1_ is the initial crack load, P_m_ is the peak load, L is the span of the specimen, b is the width of the specimen, h is the thickness of specimen, δ_1_ is the deflection corresponding to the initial crack load, δ_m_ is the deflection corresponding to the peak load, and δ is the deflection corresponding to two-thirds of the initial crack load.

## 3. Results and Discussions

### 3.1. Test Results

Based on the load-deflection data collected from the four-point bending tests, the initial crack and peak strengths, yield length, and flexural modulus of elasticity of the 16 groups of specimens were calculated, and the results are illustrated in Figure 6, Figure 7 and Figure 8, respectively.

As shown in Figure 6, during the whole bending test, the initial crack strength of SSPP-ECC was 4.67–6.93 MPa and the peak strength was 5.60–8.07 MPa. After the initial crack strength of SSPP-ECC was reached, the strength would continue to increase to the peak strength with a significant increase in deflection, which is called the deformation hardening characteristic, though the growth rate (about 20% on average) was smaller than that of classical PVA-ECC. It should be pointed out that the strengths of groups 4 and 8 were relatively low, whose mix proportions adopted a high level of water–binder ratio in the factorial design; that is to say, the strength robustness of SSPP-ECC may be reduced by an excessive water–binder ratio.

It can be seen from Figure 7 that the yield length of SSPP-ECC varied significantly with the change of the mix proportion, where groups 10, 12, 14, and 16 showed an obvious ductility advantage. In contrast, the yield length of groups 3 and 4 were much smaller, whose mix proportions adopted a high sand–binder ratio and a low level of fly ash in the factorial design.

As can be observed in Figure 8, the flexural moduli of elasticity of SSPP-ECC were mainly between 25–32 GPa. As a whole, the SSPP-ECC specimens with a high sand–binder ratio tended to show larger bending moduli. It should be noted that the above test values indicated that SSPP-ECC had lower deformability than classical PVA-ECC [1,4] but it still displays the material characteristics of obvious deformation hardening, which would be popular for many engineering concrete construction projects, especially when economical factors need to be considered.

### 3.2. Factor Analysis

In this paper, the factors A, B, C, and D were used as the parameters, and the initial crack strength, yield length, and flexural moduli of elasticity were used as the responses for the factor analysis. The F-test method from mathematical statistics was used to analyze the significance of each parameter to the responses. To show the analysis results intuitively, Pareto charts were used to show the effect value of each parameter corresponding to each response. The larger the effect value, the more significant the effect of the parameter on the response. Figure 9, Figure 10 and Figure 11 display the Pareto charts of the initial crack strength, yield length, and flexural moduli of elasticity of 16 groups of SSPP-ECC specimens. Then, the main effects of each response were selected, whose values are provided in Table 5.

It can be seen from Figure 9 that A and B were the main effects of the initial crack strength, i.e., the amount of fiber and fly ash had a significant effect on the initial crack strength of SSPP-ECC. By comparison, the effects of the sand–binder and water–binder ratios on the initial crack strength were weak within the scope of this study.

As seen in Figure 10 and Table 5, there were three main effects on the yield length of SSPP-ECC, with a significance relationship of A > B > AB, which means that both the fiber and fly ash had positive effects on the yield length and they had an extra combined effect on the yield length as well. In other words, the combination of fiber and fly ash had an impressive impact on the ductility of SSPP-ECC, which indirectly reflected the importance of fly ash to the balance between the fiber and matrix. The fibers distributed in random directions spanned the pore structure inside the matrix and played the role of transferring the load, while the fly ash played the role of micro-aggregate filling, effectively reducing the size of internal defects and making cracks finer and more uniform. When the specimen was initially cracked, the pull and energy dissipation of fibers restricted the rapid expansion of the crack and transferred the load to cause more microcracks nearby. As a result of the total of distributed deformation from each microcrack, the strength and ductility of the SSPP-ECC improved.

Figure 11 demonstrates the effects on the flexural modulus of elasticity of SSPP-ECC with a significance relationship of D > B > A. Combined with the results in Table 5, it was revealed that the flexural modulus increased with the increase of the sand–binder ratio but decreased with the increase of the content of fly ash. 

According to the significance analysis results of the factors, there were main effects that significantly determined the values of each response, whose relationships may be described quantitatively. The equations for each response were fitted using Minitab Statistical Software (Minitab, LLC, PA, USA) based on the test data of 16 groups of specimens, as illustrated in Table 6.

To evaluate the fitness of the regression models, the multiple correlation coefficient (R^2^) and adjusted multiple correlation coefficient (R^2^_adj_) were calculated and are presented in Table 6. The R^2^ represents the regression model error as a percentage of total error, where the larger the R^2^ is, the better the fitness is. R^2^_adj_ is used to further reflect the reliability of the fitting equations, where the closer R^2^_adj_ is to R^2^, the more reliable the regression model is. The results show that R^2^ for all responses were greater than 90%, and R^2^_adj_ for all responses were greater than 70%, which means the models were correlated with the actual situation. However, the R^2^_adj_ for the initial crack strength and flexural modulus of elasticity were lower than 85%, which means the prediction accuracy of the model would be limited. The yield length is an intuitive representation of the ductility, which is one of the most important indexes of SSPP-ECC. In this paper, R^2^ and R^2^_adj_ for yield length were 99.42% and 98.26%, respectively, which are very close to 1. This means that the accuracy of the regression model was quite high in the ranges of the factors, which could be an efficient tool for the design of SSPP-ECC. Figure 10, Table 5 and Table 6 revealed that factors A and B were almost completely determined the yield length of SSPP-ECC. Therefore, it would also be useful and convenient to predict the yield length of SSPP-ECC through the response surface model of factors A and B, as shown in Figure 12.

### 3.3. Crack-Control Ability

It is generally known that there is a multiple-cracking phenomenon throughout the deformation and hardening stages of ECC. Figure 13 shows the load-deflection curves of groups 8, 9, 10, 12, 14, and 16, in which the deformations of the specimens of groups 8 and 9 were far behind the others, mainly due to a lower fiber content in their mix proportions. It is hard to see the difference between groups 10, 12, 14, and 16 on the load-deflection curves. In addition to the deformation ductility, the crack-control ability of ECC is also important. In this study, the numbers of cracks produced in the specimens of the 16 groups of SSPP-ECC were different during the four-point bending test. As shown in Figure 14, more cracks appeared in specimens of groups 10, 12, 14, and 16, while other specimens cracked similar to the specimens of groups 8 or 9. However, it is not rigorous to evaluate the crack resistance of SSPP-ECC only by considering the number of cracks. If the crack width is large enough, the later deformation of ECC will be meaningless. Therefore, the maximum crack width during the deformation process should be a more important index.

It is generally considered that if the crack width is controlled to under 100 μm, it belongs to the category of harmless crack [28,29]. The maximum crack width–deflection curve of SSPP-ECC is shown in Figure 15, which describes the variation of the maximum crack width in the process of the initial crack deflection, 100 μm crack deflection, and peak deflection. The following can be concluded:With the increase of the deflection, the crack width grew increasingly faster.All specimens had similar initial crack deflections but the corresponding initial crack widths were different with the following order of sizes: C8 > C16 > C14 > C9> C12 > C10.When the maximum crack widths reached 100 μm, the deflection growths of each group were different, where the order was C10 > C12 > C14 > C16 > C9 > C8, in which the deflection growths of groups 10 and 12 were significantly larger than that of other groups.

Combined with the mix proportions shown in Table 4, it could be concluded that the precondition to ensure the crack-control ability of SSPP-ECC is to have a reasonable yield length first; otherwise, the crack width will be worse, as seen in specimens of groups 8 and 9. According to Section 3.2, factors A and B should be correctly tailored first. Then, the rule is that the larger the sand–binder ratio, the larger the crack width, which explains the phenomenon where the crack widths of groups 8, 14, and 16 were always larger than the others. For example, the yield lengths of groups 10 and 14 were quite close but the deformation growth experienced by group 10 was much larger than that of group 14 when the maximum crack width reached 100 μm, which was due to the large sand–binder ratio of group 14. This also applies to groups 12 and 16.

Based on the above analysis, summarizing the influence of mix proportion on the ductility and crack-control ability of SSPP-ECC, the contents of fiber (factor A), fly ash (factor B), and superfine sand (factor D) had significant effects on the performance of SSPP-ECC in the parameter ranges of this study. More specifically, fiber and fly ash mainly determined the level of ductility, and the ductility was positively related to the crack width. It should also be noted that a high content of superfine sand led to an increased crack width. That is to say, the mix proportions of groups 10 or 12 are recommended for SSPP-ECC based on the results of this study.

## 4. Conclusions

SSPP-ECC is a new kind of cement-based material with an engineering affinity and is economical and sustainable. In this study, a factorial design/factor analysis method was applied to study the flexural strength, flexural modulus of elasticity, ductility, and crack-control ability of SSPP-ECC. The main conclusions are as follows:The initial flexural crack strength of SSPP-ECC was 4.67–6.93 MPa, the peak flexural strength was 5.60–8.07 MPa, and the flexural moduli of elasticity of SSPP-ECC were mainly between 25 and 32 GPa, in which the content of fiber and fly ash had a significant effect on the initial crack strength, while the flexural modulus of elasticity was greatly affected by the content of fiber and the sand-binder ratio.The content of fiber and fly ash almost completely determined the ductility of SSPP-ECC in the parameter ranges of this study, and they had an extra combined effect on the ductility as well. Furthermore, a response surface model with high accuracy was fitted in this paper to predict the yield length of SSPP-ECC.The ductility of SSPP-ECC was positively related to the crack-control ability and it was shown that the crack width of SSPP-ECC increased significantly with a high content of superfine sand. Finally, the mix proportions of groups 10 and 12 were recommended for SSPP-ECC based on the results of this study.

In this study, the SSPP-ECC achieved far more deflection than ordinary concrete materials while controlling the maximum crack width to be no more than 100 μm, which could easily cater to the demands of many concrete structures, such as a bridge deck wet-joint, protective layer of reinforced concrete, and so on. In terms of sustainable development, deposited superfine sand can be utilized in sustainable and economical ways by producing SSPP-ECC for practical constructions. Additionally, further studies will be based on plenty of experiments at multiple levels with repeated observations, including the optimal mix proportions for different engineering applications, durability tests, practical applications in bridge structures, and so on.

## Figures and Tables

**Figure 1 materials-13-02609-f001:**
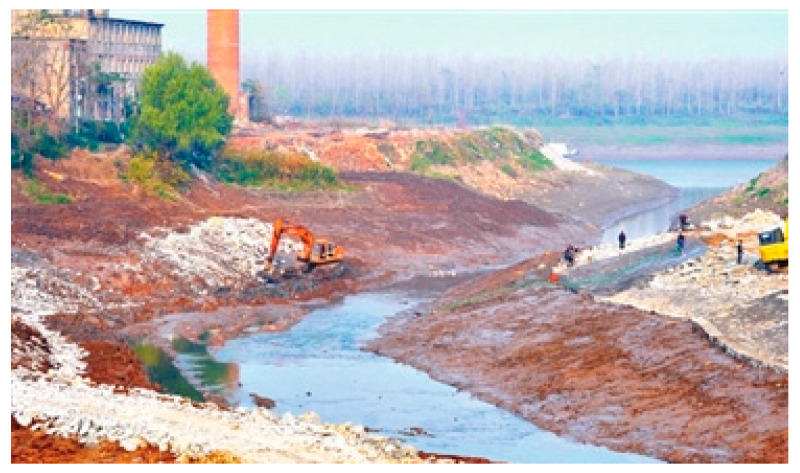
A site of superfine river sand silted up causing a river blockage in China.

**Figure 4 materials-13-02609-f004:**
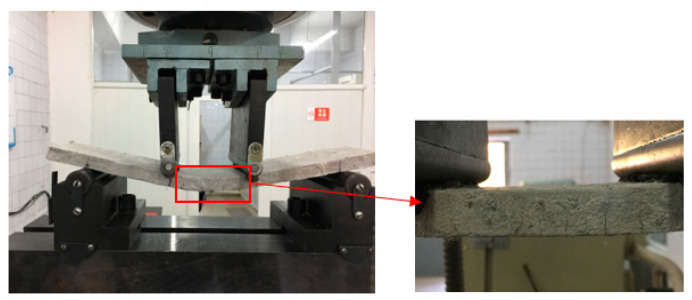
Four-point bending test.

**Figure 5 materials-13-02609-f005:**
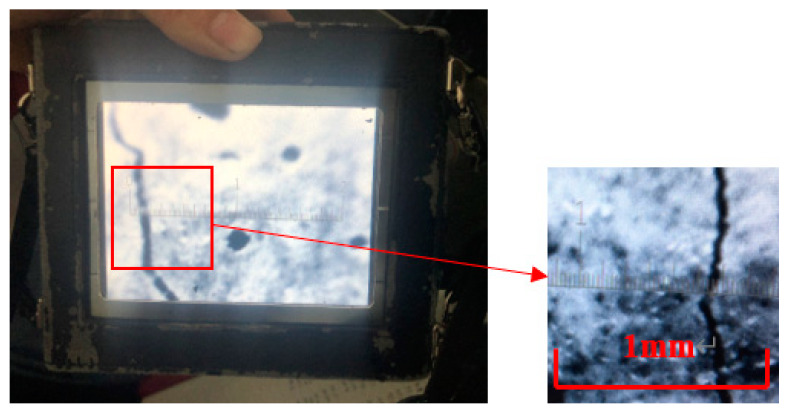
Crack width measurement.

**Figure 6 materials-13-02609-f006:**
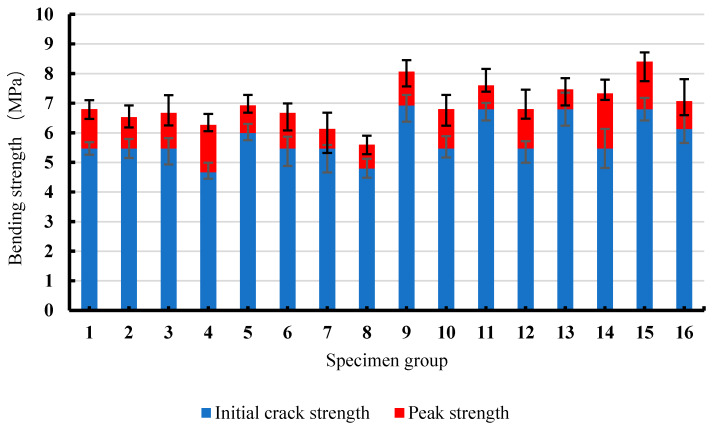
The initial crack and peak strengths of 16 groups of specimens.

**Figure 7 materials-13-02609-f007:**
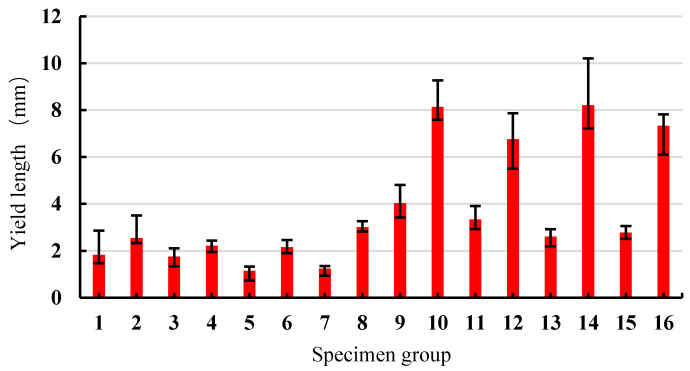
The yield lengths of 16 groups of specimens.

**Figure 8 materials-13-02609-f008:**
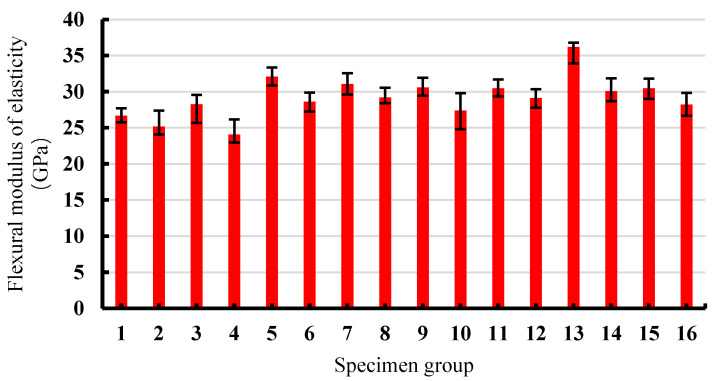
The flexural moduli of elasticity of 16 groups of specimens.

**Figure 9 materials-13-02609-f009:**
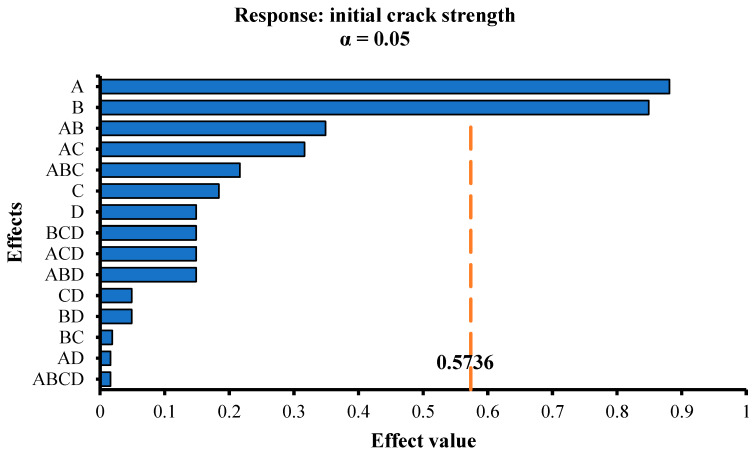
Pareto chart of the initial crack strength.

**Figure 10 materials-13-02609-f010:**
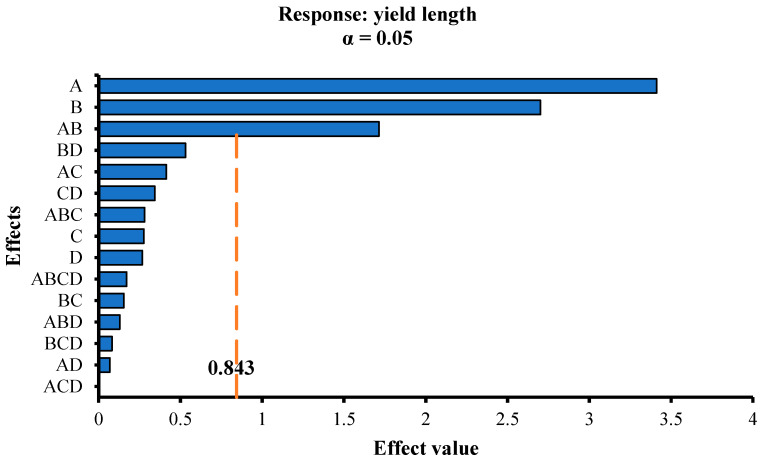
Pareto chart of the yield length.

**Figure 11 materials-13-02609-f011:**
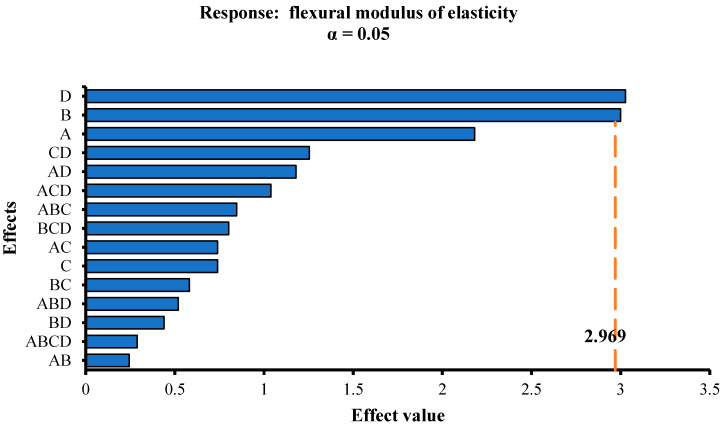
Pareto chart of the flexural modulus of elasticity.

**Figure 12 materials-13-02609-f012:**
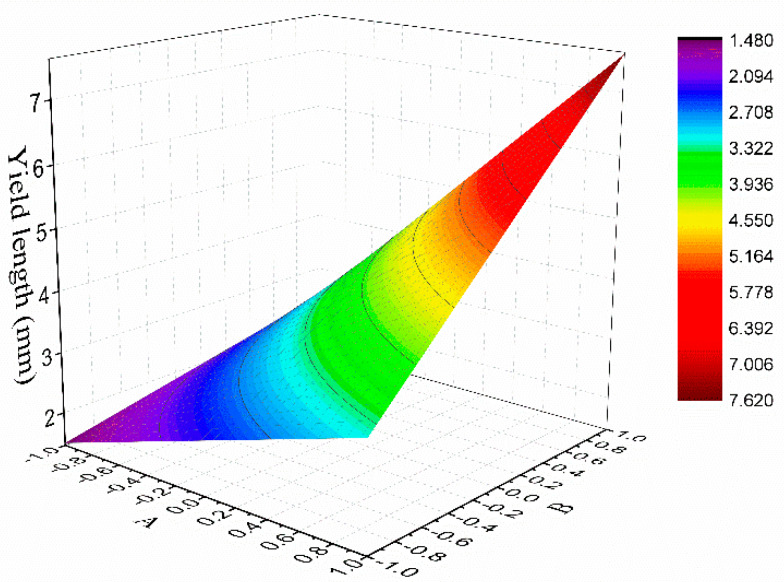
The response surface model for yield length.

**Figure 13 materials-13-02609-f013:**
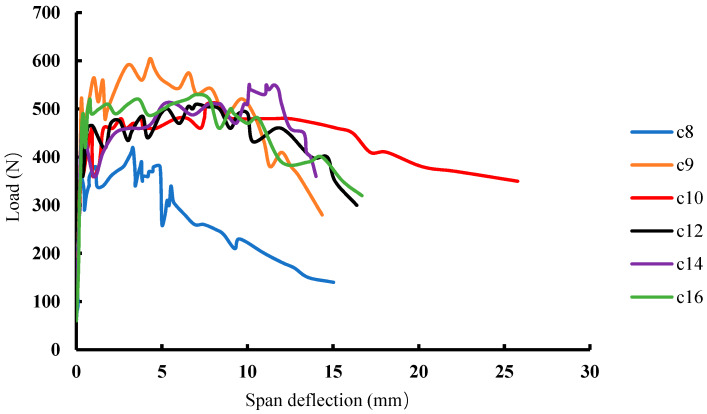
The load-deflection curves of SSPP-ECC.

**Figure 14 materials-13-02609-f014:**
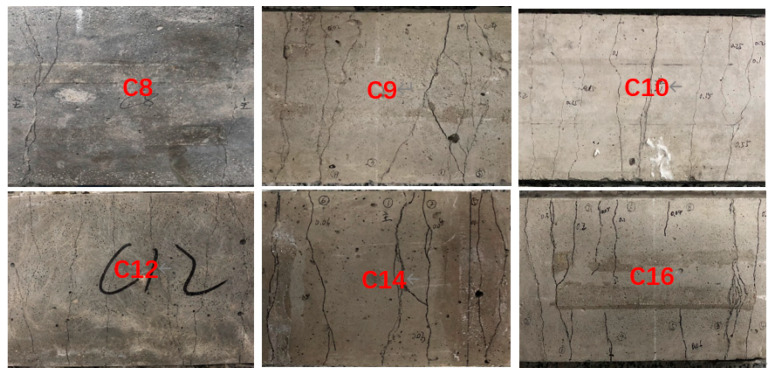
Typical cracking states of SSPP-ECC.

**Figure 15 materials-13-02609-f015:**
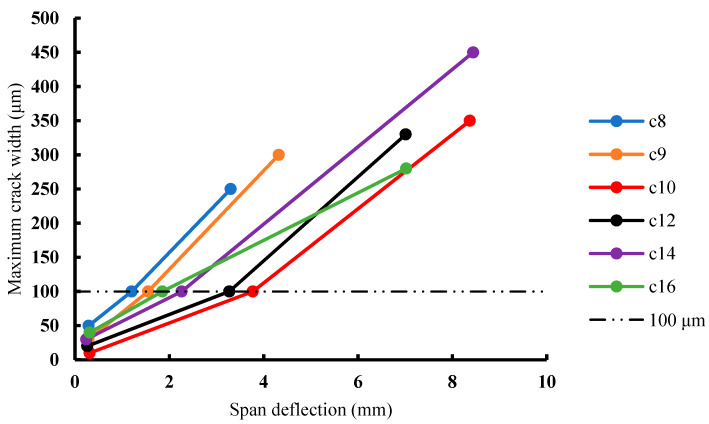
The maximum crack width–deflection curves of SSPP-ECC.

**Figure 2 materials-13-02609-f002:**
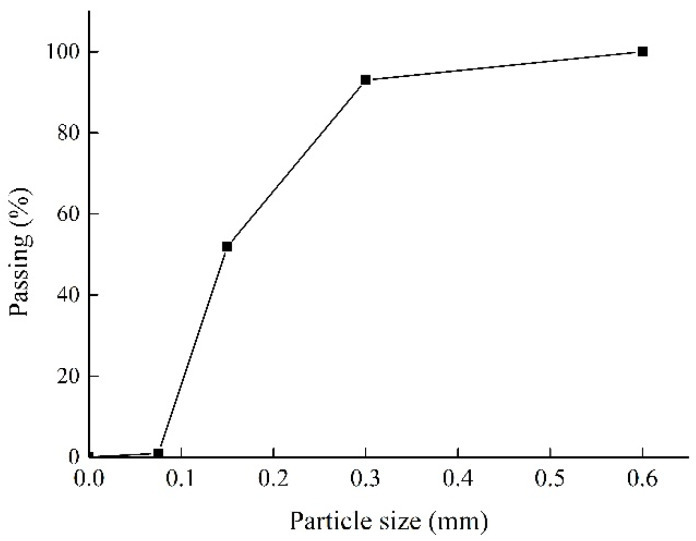
Grading curve of the superfine sand.

**Figure 3 materials-13-02609-f003:**
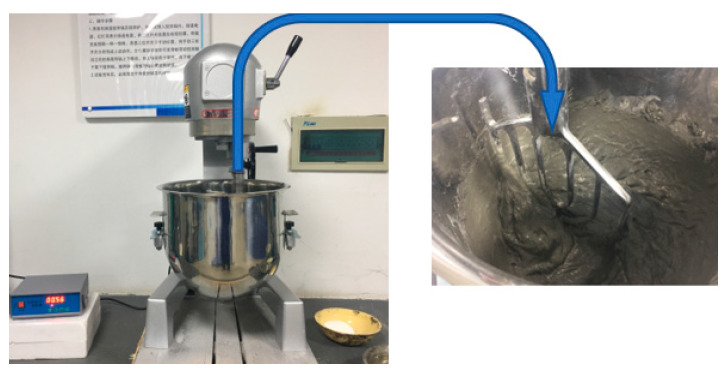
The state of fresh engineered cementitious composite with superfine river sand and polypropylene fiber (SSPP-ECC) after the mixing process.

**Table 1 materials-13-02609-t001:** Main components and properties of cementitious binders.

Properties	Fly Ash	Cement
Specific gravity	2.13	3.10
Surface area ratio (m^2^/kg)	420	370
SiO_2_ (%)	50.37	21.11
CaO (%)	3.01	60.38
Al_2_O_3_ (%)	27.62	6.04
MgO (%)	1.85	1.08
Fe_2_O_3_ (%)	7.83	2.56
Loss on ignition (%)	7.20	1.02
Water ratio (%)	0.80	0.11

**Table 3 materials-13-02609-t003:** The levels of factorial design.

Factors	Low (-)	High (+)
A (volume content of fiber, %)	1.5	2.0
B (percentage of fly ash, %)	20	60
C (water–binder ratio)	0.3	0.35
D (sand–binder ratio)	0.5	1.0

**Table 4 materials-13-02609-t004:** Mix proportions of SSPP-ECC.

No.	Cement (kg/m^3^)	Fly Ash (kg/m^3^)	Water (kg/m^3^)	Sand (kg/m^3^)	HRWR (kg/m^3^)	Fiber (%)
1	960	240	360	600	11.3	1.5
2	480	720	360	600	5.7	1.5
3	960	240	420	600	11.4	1.5
4	480	720	420	600	5.7	1.5
5	960	240	360	1200	11.5	1.5
6	480	720	360	1200	5.6	1.5
7	960	240	420	1200	11.4	1.5
8	480	720	420	1200	5.7	1.5
9	960	240	360	600	11.3	2.0
10	480	720	360	600	5.6	2.0
11	960	240	420	600	11.4	2.0
12	480	720	420	600	5.7	2.0
13	960	240	360	1200	11.5	2.0
14	480	720	360	1200	5.3	2.0
15	960	240	420	1200	11.4	2.0
16	480	720	420	1200	5.4	2.0

HRWR: High-Range Water Reducer.

**Table 5 materials-13-02609-t005:** The values of the main effects of each response.

Initial Crack Strength	Yield Length	Flexural Modulus of Elasticity
Inspection value: 0.5736 A: +0.8812 B: ‒0.8488	Inspection value: 0.843 A: +3.411 B: +2.701 AB: +1.7137	Inspection value: 2.969 D: +3.026 B: ‒2.999
–		–

The (+) and (−) signs represent positive correlation and negative correlation respectively.

**Table 6 materials-13-02609-t006:** Fitting equations of each response.

Responses	Fitting Equations	R^2^ (%)	R^2^_adj_ (%)
Initial crack strength	Y_I_ = 5.7931 + 0.4406A − 0.4244B − 0.0919C + 0.0744D − 0.1744AB + 0.1581AC − 0.0081AD − 0.0094BC + 0.0244BD + 0.0244CD	94.01	82.03
Yield length	Y_Y_ = 3.6894 + 1.7056A + 1.3506B − 0.1381C − 0.1331D + 0.8569AB − 0.2069AC − 0.0344AD − 0.0769BC + 0.2656BD + 0.1719CD	99.42	98.26
Flexural modulus of elasticity	Y_F_ = 29.231 + 1.091A − 1.499B − 0.3694C + 1.513D − 0.1219AB − 0.3694AC − 0.5894AD + 0.2906BC − 0.2194BD − 0.6269CD	90.81	72.42

**Table 2 materials-13-02609-t002:** Properties of polypropylene (PP) fiber.

Diameter (μm)	Length (mm)	Density (g/cm^3^)	Young’s Modulus (GPa)	Nominal Strength (MPa)	Elongation (%)
30	12	0.91	3.5	500	20

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
