# Peer review of "Preliminary Analysis of the Ductility and Crack-Control Ability of Engineered Cementitious Composite with Superfine Sand and Polypropylene Fiber (SSPP-ECC)"

_materials, 2020, doi:10.3390/ma13112609_

Round 1

Reviewer 1 Report

Minor spell check required, for example: line 34 "are concentrate";

It is recommended to calculate the measurement uncertainty concerning the bending strength, deflection, flexural modulus of elasticity, crack width measured during the whole bending tests.

Author Response

Point 1: Minor spell check required, for example: line 34 "are concentrate". 

Response 1: Thanks very much for your valuable comment. The language, spelling mistakes and sentences throughout the whole manuscript have been checked carefully. Some changes are listed as follows:

Line 41 "are concentrate" is changed into "are concentrated".

Line 148 "were disposed follow" is changed into "were disposed following".

Point 2: It is recommended to calculate the measurement uncertainty concerning the bending strength, deflection, flexural modulus of elasticity, crack width measured during the whole bending tests.

Response 2: Thanks very much for your instructive comments. This work conducted repeated experiments and the error bars of bending strength, deflection and flexural modulus of elasticity have been added in Figures 6-8. Actually, we have prepared a large number of specimens for the further research. Combining your rigorous suggestions, the control specimens and measurement uncertainty will be scientifically set up in the following experiments at multiple levels with repeated observations including performances in various structural forms, durability under environment and load, the optimum mix proportions for different engineering applications and so on.

Reviewer 2 Report

In this study, the major parameters affecting the quality of the Engineered Cementitious Composites with Superfine Sand and Polypropylene Fiber (SSPP-ECC) were experimentally and statistically analyzed.

It is scientifically worth doing it to clearly see the engineering boundaries of these composites in an ideal way and adjusting the properties according to the mechanical needs.

In terms of experimental design and statistical analysis, it is a clear and modest study. The parameters they checked, results, and the analyses were nicely presented and the conclusions are in correlation with them.

The study both helps to develop the composites and also guides further research in this field. 

It might be published as it is.

Author Response

Point 1: In this study, the major parameters affecting the quality of the Engineered Cementitious Composites with Superfine Sand and Polypropylene Fiber (SSPP-ECC) were experimentally and statistically analyzed.

It is scientifically worth doing it to clearly see the engineering boundaries of these composites in an ideal way and adjusting the properties according to the mechanical needs.

In terms of experimental design and statistical analysis, it is a clear and modest study. The parameters they checked, results, and the analyses were nicely presented and the conclusions are in correlation with them.

The study both helps to develop the composites and also guides further research in this field.

It might be published as it is.

ʉ۬

Response 1: We are grateful for your recognition of our paper, and we have carefully proofread the logic and spelling of the whole manuscript.

Reviewer 3 Report

The authors present a research work related to “Preliminary Analysis on the Ductility and Crack Control Ability of Engineered Cementitious Composite with Superfine Sand and Polypropylene Fiber (SSPP-ECC)” where a factorial design – analysis method was applied to study the flexural strength, flexural modulus of elasticity, ductility and crack control ability of SSPP-ECC.

Remarks to the authors:

  1. In the Materials and Methods section please specify if the PP fibers were uniformly distributed.
  2. Please mention if there is any control specimen used in the Results analysis section.
  3. Please relate how your results can be interpreted in the context of previous studies.
  4. Insert the standards used in the experimental part.
  5. Future research directions may also be mentioned.

Author Response

Point 1: In the Materials and Methods section please specify if the PP fibers were uniformly distributed. 

Response 1: Thanks very much to point out this important modification. It is very essential to emphasize the uniform dispersion of fibers. The detailed modification is listed as follows:

Line 151 "remaining water were added and mixed well" is changed into "remaining water were added and the mixture was stirred for at least 5 minutes to ensure uniform fiber distribution".

Point 2: Please mention if there is any control specimen used in the Results analysis section.

Response 2: Thanks very much for your valuable comment. This paper is focused on the influence of the mix proportion on the performance of SSPP-ECC based on a 24 factorial design method, and no separate control specimen was prepared. Actually, we have prepared a large number of specimens for the further research. Combining your rigorous suggestions, the control specimens and measurement uncertainty will be scientifically set up in the following experiments at multiple levels with repeated observations including performances in various structural forms, durability under environment and load, the optimum mix proportions for different engineering applications and so on.

Point 3: Please relate how your results can be interpreted in the context of previous studies.

Response 3: Thanks very much for your constructive comment. We have added the comparison of SSPP-ECC to classical PVA-ECC by referring to previous literature review. The detailed modifications are listed as follows:

Line 233 "It should be noted that above test values indicated that SSPP-ECC is less than classical PVA-ECC in deformability [1,4], but it still identifies the material characteristics of obvious deformation hardening, which would be popular for many engineering concrete constructions, especially when economy needs to be considered."

Point 4: Insert the standards used in the experimental part.

Response 4: Thanks very much to point out this important modification. The detailed modification is listed as follows:

Line 181 "for four points bending as per Chinese specification GB/T 15231-2008 [29]"

Line 200 "by Equations (1)- (4) according to Chinese specification GB/T 15231-2008 [29]."

Point 5: Future research directions may also be mentioned.

Response 5: Thanks very much for your constructive comment. We have added the future research in Line 367:

"In addition, the further more researches in progress will be based on plenty of experiments at multiple levels with repeated observations, including the optimum mix proportions for different engineering applications, durability tests, practical applications in bridge structures and so on."

Reviewer 4 Report

Abstract

I suggest to the authors do not use acronyms in the abstract to facilitate the reader. Moreover a short state of art is completely missing and the authors need to provide it.

Introduction

I think that the authors should combine introduction with background section because the two parts are linkable and easily readable together

Materials and Methods

Pag 5 line 153 the authors need to indicate the load cell that they have been used, the specimens dimensions, the number of specimens that they have tested and if they have follow a standard

Pag 5 line 155 “At the same time, the crack widths of specimens in different states were collected by using the crack observation instrument” which instrument? what resolution does it have? authors must give more information.

Author Response

Point 1: Abstract

I suggest to the authors do not use acronyms in the abstract to facilitate the reader. Moreover a short state of art is completely missing and the authors need to provide it. 

Response 1: Thanks very much for your valuable comment. We polished the organization of our ‘Abstract’ based on your comments. Corresponding changes are listed as follows:

Line 13 to line 30 Abstract: The engineered cementitious composite (ECC) is a potential cement-based material with abilities of large deformation and crack width control. However, ECC is difficult to be popularized in many developing countries, because the costs of silica sand and PVA fiber with surface coating are too high for practical engineering. Therefore, the authors proposed the economical ECC with superfine river sand and polypropylene fiber (SSPP-ECC) replacing PVA fiber and silica sand. The SSPP-ECC proposed in this paper is a sustainable material using local material ingredients, which has considerable adaptability for large-scale engineering applications. The 16 groups of specimens were prepared through factorial design method, curing for four points bending tests. The bending strength, deflection, flexural modulus of elasticity and crack width were measured and calculated during the test. The factor analysis of test results shows that the contents of fiber and fly ash have significant effects on the ductility of SSPP-ECC with an extra combined effect at the same time, and a response surface model with high accuracy was fitted to predict the yield length of SSPP-ECC. The ductility of SSPP-ECC is positively related to its crack control ability, and it was proved that the crack width of SSPP-ECC increases significantly with a high content of superfine sand. This paper proposed a reasonable way to utilize superfine sand, and provided the mix proportion of SSPP-ECC with characteristics of deformation hardening and multi-cracking, which may cater to demands of many concrete components on ductility and crack resistance.

Point 2: Introduction

I think that the authors should combine introduction with background section because the two parts are linkable and easily readable together.

Response 2: Thanks very much for your constructive comment. It is more reasonable to combine the introduction with background section, and corresponding changes are listed in the revised version in line 78-135.

Point 3: Materials and Methods

Page 5 line 153 the authors need to indicate the load cell that they have been used, the specimens dimensions, the number of specimens that they have tested and if they have follow a standard.

Response 3: Thanks very much to point out this important modification. Corresponding changes are listed as follows:

Line 180 to line 184 “Based on the 24 factorial design, 16 groups of specimens were prepared according to the mix proportions in Table 4. After curing for 28 days, the specimens with dimensions of 15mm × 100mm × 400mm were tested with mechanical testing and simulation (MTS) device for four points bending as per Chinese specification GB/T 15231-2008 [29], using strain-control condition at an axial strain rate of 0.2 mm/min, as shown in Figure 4. MTS carried load cell with force measurement of 0-10 KN and its system recorded the changes of load and deflection in the whole process.”

Point 4: Page 5 line 155 “At the same time, the crack widths of specimens in different states were collected by using the crack observation instrument” which instrument? what resolution does it have? authors must give more information.

Response 4: Thanks very much to point out this important modification. Corresponding changes are listed as follows:

Line 185 “At the same time, the crack widths of specimens in different states were collected by using the ZBL-F103 crack observation instrument produced by Beijing ZBL Science & Technology Co. Ltd, whose measurement accuracy is 10 μm, as exhibited in Figure 5.”
